# Metal-Assisted Chemical Etching for Anisotropic Deep Trenching of GaN Array

**DOI:** 10.3390/nano11123179

**Published:** 2021-11-24

**Authors:** Qi Wang, Kehong Zhou, Shuai Zhao, Wen Yang, Hongsheng Zhang, Wensheng Yan, Yi Huang, Guodong Yuan

**Affiliations:** 1School of Optoelectronic Engineering, Chongqing University of Posts and Telecommunications, Chongqing 400065, China; wangqi@cqupt.edu.cn (Q.W.); S190431014@stu.cqupy.edu.cn (K.Z.); S200402033@stu.cqupt.edu.cn (W.Y.); zhanghs@cqupt.edu.cn (H.Z.); yws118@163.com (W.Y.); 2State Key Laboratory for Superlattices and Microstructures, Institute of Semiconductors, Chinese Academy of Sciences, Beijing 100083, China; szhao@semi.ac.cn; 3College of Materials Science and Opto-Electronic Technology, University of Chinese Academy of Sciences, Beijing 100049, China

**Keywords:** wet etching, MacEtch, GaN, deep trenches, microstructures

## Abstract

Realizing the anisotropic deep trenching of GaN without surface damage is essential for the fabrication of GaN-based devices. However, traditional dry etching technologies introduce irreversible damage to GaN and degrade the performance of the device. In this paper, we demonstrate a damage-free, rapid metal-assisted chemical etching (MacEtch) method and perform an anisotropic, deep trenching of a GaN array. Regular GaN microarrays are fabricated based on the proposed method, in which CuSO_4_ and HF are adopted as etchants while ultraviolet light and Ni/Ag mask are applied to catalyze the etching process of GaN, reaching an etching rate of 100 nm/min. We comprehensively explore the etching mechanism by adopting three different patterns, comparing a Ni/Ag mask with a SiN mask, and adjusting the etchant proportion. Under the catalytic role of Ni/Ag, the GaN etching rate nearby the metal mask is much faster than that of other parts, which contributes to the formation of deep trenches. Furthermore, an optimized etchant is studied to restrain the disorder accumulation of excessive Cu particles and guarantee a continuous etching result. Notably, our work presents a novel low-cost MacEtch method to achieve GaN deep etching at room temperature, which may promote the evolution of GaN-based device fabrication.

## 1. Introduction

Metal-assisted chemical etching (MacEtch) is crucial for preparing complex micro-/nano-structures such as pores, rings and pillars on Si and other compound semiconductors [1,2]. At present, Si/Ge etching is the focus, and the progress of Si is much more mature [3]. The etching of III-V compounds (GaAs/InP/GaN) has also been carried out [4,5,6], but the etching mechanism of wide-bandgap semiconductors is imperfect and needs to be explored further. Unlike the traditional Si/Ge etching, the III-V semiconductor compounds have different characteristics. Additionally, the corresponding etching morphology and mechanism are not the same for different compounds. Etching conditions (assisted by temperature, electricity, and ultraviolet light) and the selection of oxidants/reductants all need to be explored [7,8,9]. The MacEtch technique has shown advantages in the etching of III-V semiconductor compounds with narrow bandgaps (such as GaAs) [10,11]. However, the research on wide-bandgap semiconductors (e.g., GaN) has not been completely established, especially for their deep etching [12,13]. Significant advantages in photoelectric devices have been shown in GaN due to its higher mobility and higher breakdown voltage [14,15,16]. Currently, GaN micro-/nano-structures are mainly prepared through bottom-up growth and top-down dry etching [17,18]. The epitaxial growth system is complex, and dry etching damages the material. Thus, a promising method, MacEtch, was studied to etch GaN micro-/nano-structures in [19,20]. This relatively simple approach does not damage GaN, which contributes to the better performance of devices. However, it is difficult to etch wide-bandgap GaN using conventional MacEtch because electron–hole pairs cannot be directly decomposed. An external force, such as a light or power source, is required to drive the anode decomposition of the semiconductor [21,22]. Though there are reports about GaN nanowires, few researchers have paid attention to the deep etching of GaN. In our previous work, GaN nanowires were obtained by applying ultraviolet light as the driving light source. Moreover, we found that the etching reaction stopped once the metal accumulated on the surface of the reaction sheet [23].

In this paper, we successfully realized uniform GaN microarrays with different depths via MacEtch using Ni/Ag as the mask. The process and reaction principle of MacEtch are presented in Section 2, while the scanning electron microscopy (SEM), atomic force microscopy (AFM) experimental results, and mechanism of MacEtch are shown in Section 3. We prove that the anisotropic, deep trenching of GaN microarrays can be rapidly prepared by the proposed MacEtch method.

## 2. Materials and Methods

The 6 µm-thick n-type GaN (nGaN) layer with a carrier concentration of 3.0 × 10^18^ cm^−3^ is grown on a 2″ sapphire substrate by metal organic chemical vapor deposition (MOCVD). First, 200 nm undoped GaN (uGaN) buffer layers are grown on the sapphire substrate, followed by the 6 µm-thick n-type GaN layer. Figure 1a–f shows the basic process flow of MacEtch for the preparation of the samples. Figure 1a shows the GaN sample, in which the scale of the sapphire substrate is adjusted for reading convenience. As shown in Figure 1b, the photoresist layer is spin coated on the GaN. Then, the photoresist is exposed and developed to generate patterns. Three different shapes are patterned on the GaN surface: micropillars with a 5 µm diameter and 5 µm gap; squares with a 50 µm side and 5 µm gap; and stripes with a 10 µm width and 3 µm gap, as shown in Figure 1c. A Ni/Ag (100/300 nm) layer is then deposited on the GaN surface by electron beam evaporation. The metal Ni/Ag is usually applied as the mask in the MacEtch method due to its corrosion resistance. The sample after Ni/Ag lift-off is shown in Figure 1e. Additionally, the GaN chip is divided into squares with a 1 cm side after the Ni/Ag is stripped and cleaned. These chips are put in the MacEtch etchant, as shown in Figure 1f. The etchant consists of a 0.01 M CuSO_4_ solution with 5 M HF and DI water. The etching time is 20 to 60 min under 300 mW ultraviolet (UV) light. Then, the chips are covered by a Cu film after the reaction, as shown in Figure 1g. The etched chips are placed in the diluted HNO_3_ solution for 10 min to remove the Cu particles and Ni/Ag mask. Finally, the etched chip shown in Figure 1h is obtained. 

The schematic diagram of the GaN MacEtch reaction is presented in Figure 2. The wide bandgap prevents GaN from being directly etched, and thus external forces such as electricity or UV light are needed to separate electron–hole pairs [24]. In this paper, UV illumination is applied to separate electron–hole pairs of GaN. The generation, transfer, and consumption of electron–hole pairs promote the continuous decomposition of GaN. As Figure 2 shows, the electrons and holes are separated under UV illumination. The valence electrons are excited from the valence band (VB) to the conduction band (CB) of GaN. Then, the excited electrons combine with Cu^2+^ to generate Cu. Under the attraction of the metal mask, the generated Cu gradually forms on its edge. At the same time, the generated holes are consumed to preferentially oxidize GaN due to the higher oxidation potentials ϕ_ox_ of GaN than the O_2_/H_2_O oxidation potential $\phi_{O_{2}/H_{2}O}$ [25,26]. Then, Ga^3+^ and HF react with each other, generating a kind of gallium fluoride [12,27]. The main reactions can be expressed as follows:

Cathode:Cu^2+^+2e^−^ → Cu(1)

Anode:2GaN + 6h^+^ → 2Ga^3+^ + N_2_ ↑(2a)
Ga^3+^ + xHF → GaF_x_
^3−x^ + xH^+^(2b)

## 3. Results and Discussion

Three different GaN trench arrays are fabricated to reveal the characteristics of our proposed MacEtch method. The SEM images of micropillar, square, and stripe arrays are shown in Figure 3, demonstrating a good deep-etching capacity. Figure 3a shows the uniform array, which consists of micropillars with a 5 µm diameter and 5 µm spacing. Nevertheless, there is a certain inclination angle between the height and the plane of the micropillar, requiring some adjustment to improve the verticality of the sidewall. Compared with the micropillars array, the squares and stripes arrays present a better performance and verticality of the sidewall, as shown in Figure 3b,c. The grooves present noticeable edges and angles, as well as a relatively high verticality of the sidewall.

The stripe grooves etched for different durations (20, 40, and 60 min) can be observed in Figure 4. Due to the high hardness of sapphire substrates, it is difficult to ensure a straight cleaved section. Thus, the groove orientations are different in SEM images to observe the regions where the cross sections are approximately vertical. The red tag in Figure 4a marks a V-type angle between the horizontal and vertical directions at the bottom of the groove. Because the catalysis of the accumulated Cu on the stripe edge speeds up the reaction, the area nearby the Ni/Ag mask is etched first; then, the V-type angle forms. The etching rate in the vertical direction is larger than that in the horizontal one, which causes the horizontal width to narrow after a period of etching. Then, the inverted trapezoidal groove, together with an obvious V-type angle, occurs after a 20 min etching, as shown in Figure 4a.

However, the angle starts to become smaller as the reaction progresses, as shown in Figure 4b. There is a thin film layer at the groove bottom in Figure 4c, which comes from the inserted uGaN buffer layer during the GaN epitaxial growth. In detail, the film is the product of the transition from the uGaN to an nGaN layer. When the etching goes through the nGaN to the uGaN layer, the etching slows due to the decreased doping concentration. The strong binding force produced by the fusion of uGaN and nGaN at the interface causes difficulty in etching the interface layer. The vertical etching starts to slow down while the speed of horizontal etching stays the same because of the film’s barrier. On this occasion, the V-type angle becomes flat and the sidewall verticality becomes higher after 40 min of etching. Finally, the uGaN buffer layer is fully etched, while the interface between uGaN and nGaN is partly etched after 60 min. In Figure 4c, the remaining partly etched interface layer between uGaN and nGaN is suspended. Furthermore, the MacEtch method can also be applied to etch uGaN. In this paper, excellent GaN etching is achieved based on the proposed method.

Notably, the cylindrical structure in Figure 3a is with a large horizontal etching range, resulting in a limited reduction in the V-type angle, which causes the micropillars to present a slightly lower sidewall verticality than other patterns. In addition, the etched sidewall appears to have a porous structure, which is a sign of lateral etching. When the reaction starts, nanopores form on the sample surface, and ultraviolet light penetrates along them, making the reaction extend deeply. However, the interaction between the sidewall and ultraviolet light is weak, giving rise to a slow reaction. This is another main reason why the etching rate in the vertical direction is larger than that in the horizontal direction. It can also be seen from Figure 4 that the vertical height reaches about 3, 4.5, and 6 µm after etching for 20, 40, and 60 min, respectively, meaning that the etching rate reaches up to ~100 nm/min; meanwhile, the obtained trench arrays are uniform. Note that this high etching rate is realized with MacEtch at room temperature, and it is believed to be further improved by raising the reaction temperature. Furthermore, this result is comparable with the photoelectrochemical (PEC) etching rate (~0.2–100 nm/min) reported in previous work [28,29].

The AFM images of the etched sample with 60 and 90 min etching are used in the qualitative analyses of sample surface roughness, as shown in Figure 5a,b. The corresponding arithmetic mean roughness R_a_ values in the 1 µm^2^ square area measurement are 7.565 nm when the etching time is 60 min, and this value is 0.480 nm when the etching time is 90 min. At the beginning of etching, holes are formed on the GaN surface, causing the roughness to increase. However, as the reaction progresses, the etched surface gradually flattens and the roughness decreases.

The principle of the proposed MacEtch method is illustrated in Figure 6, which is an electrochemical reaction when the GaN is etched. It is well established that as-generated electron–hole pairs in GaN separate and diffuse in the semiconductor under ultraviolet light irradiation. Then, the holes directly oxidize GaN to generate Ga_2_O_3_ and the Ga_2_O_3_ is dissolved under the action of HF, realizing the GaN etching. At the same time, the Cu ions obtain electrons to produce Cu particles which accumulate on the metal-mask surfaces because of the attraction. The Cu particles begin to deposit around the mask edge, which catalyze the reaction nearby the mask, giving rise to the difference between vertical and lateral etching rates. This is the reason why V-type angles form at the beginning. Though anisotropic etching of GaN is realized in this paper, more optimizations are needed to further improve the verticality of the sidewall based on this theory.

### 3.1. Selection of Mask Materials

The catalysis of metal is a key element in the MacEtch process. To explore the function of the Ni/Ag mask, we introduce another nonmetallic material (SiN) as the etching mask for comparison. Figure 7a,b shows the GaN microstructures etched for 40 min with a 100 nm/300 nm Ni/Ag mask, and Figure 7c,d presents the GaN microstructures etched for 60 min with a 400 nm SiN mask. It can be observed in Figure 7a,b that a large amount of reaction-generated particles appear and pile up on the Ni/Ag mask surface (especially at the mask edge), while particles in other areas are scarce. These particles are proved to be Cu, and this phenomenon may be related to the metal mask’s attraction to copper ions. The Cu particles preferentially nucleate at these sites. As for the etched sample using a SiN mask (see Figure 7c,d), there are no massive Cu particles on the SiN mask surface. Furthermore, many smaller GaN micropillars occur in the maskless area. It can be inferred that the nonmetallic SiN mask possesses a poor ability to attract Cu^2+^, and thus the reaction-generated Cu particles preferentially nucleate and grow at other regions (defective sites). These Cu particles, serving as small irregularly shaped metal masks, effectively block the UV illumination at certain regions during the MacEtch process, inducing the formation of small GaN micropillars. However, such Cu particles may be spontaneously lifted off after the whole process due to the weak metal–semiconductor bonding force, and there are few Cu particles that can be observed after etching. Under this circumstance, we find that the randomly distributed GaN micropillars can be effectively avoided by adopting the Ni/Ag mask during MacEtch, and this metal mask is more conducive to the fabrication of regular GaN trench arrays.

### 3.2. Effect of HF Concentration on GaN MacEtch

The HF concentration plays another key role in GaN MacEtch. The as-prepared GaN micropillar arrays with two different HF concentrations are studied in this section. The etching time is 30 min and the etching mask is Ni/Ag. Figure 8a,c shows the SEM images (45°-tilt view) of the etched GaN micropillar arrays with 5 M and 10 M HF concentrations, respectively. Figure 8b,d presents the magnified SEM images of Figure 8a,c. Counterintuitively, the GaN pillar height obtained from the etchant containing 5 M HF is higher than that obtained with 10 M HF. The Cu particle distributions can also be observed in Figure 8. As mentioned, Cu particles accumulate only on the mask surface after the sample is etched in a 5 M HF-containing etchant. As for the 10 M HF-processed sample, Cu particles heap up at the mask surface and at the micropillar sidewall.

A schematic diagram is depicted and shown in Figure 9 to further reveal the role of the HF concentration on GaN etching. The orange dots represent Cu particles while the yellow bars represent the Ni/Ag masks. It can be observed from Figure 9a (corresponding to Figure 8b) that the generated Cu particles mainly accumulate on the edge and the surface of the mask. Additionally, the generated Cu particles spread over the whole sample surface including the pillar sidewalls after etching, as shown in Figure 9b (corresponding to Figure 8d). The increasing HF concentration accelerates the decomposition of GaN and the consumption of the holes (anode reaction). Accordingly, the cathodic Cu generation reaction rate is also elevated, and thus many more Cu particles come into being and spread over the whole wafer. These excessive Cu particles wrap the whole pillar sidewalls and cover the bottom maskless regions, blocking the UV light penetration and thus reducing the etching rate. In this situation, the obtained micropillars have a short structure. Therefore, it is vital to adopt a specific etchant in order to ensure the etching rate and etched morphology, and in our work the optimized CuSO_4_/HF molar ratio is 0.02:5.

## 4. Conclusions

We demonstrate a novel MacEtch method and realize the rapid and anisotropic deep etching of GaN. Regular GaN microarrays, including micropillar, square, and stripe arrays, are fabricated by this method, and the etching rate can reach 100 nm/min. Moreover, the characteristics and mechanism of the MacEtch are explored through analyzing the effects of the mask material, UV illumination, and etchant proportion. The catalytic action of the Ni/Ag mask affects the etching rate and benefits the formation of deep trenches. The UV light drives the separation of electron–hole pairs in GaN, and the specific etchant with an optimized CuSO_4_/HF ratio may ensure the orderly accumulation of Cu particles, resulting in a continuous GaN trenching. Additionally, the blocking effect of the uGaN/nGaN combination layer contributes to a high sidewall verticality. This work offers a low-cost, rapid, anisotropic deep-etching technique for the preparation of GaN microarrays at room temperature, bringing a new promise for potential GaN device fabrication. Meanwhile, we provide a new physical insight into GaN deep etching, paving a new way for future wide-bandgap semiconductor etching.

## Figures and Tables

**Figure 1 nanomaterials-11-03179-f001:**
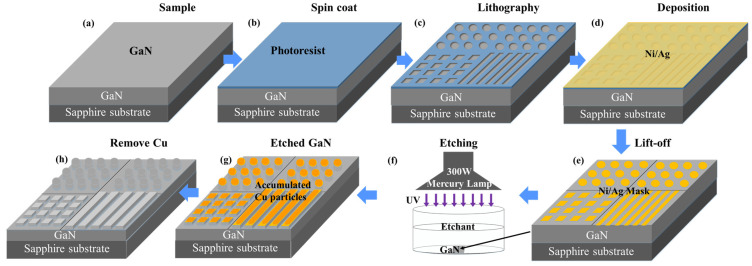
Schematic diagram showing the main steps of process flow in the fabrication of GaN: (**a**) GaN sample, (**b**) GaN coated by photoresist, (**c**) laser interference lithography, (**d**) Ni/Ag deposition, (**e**) Ni/Ag lift-off, (**f**) metal-assisted photochemical etching, (**g**) MacEtch of GaN and accumulation of Cu on Ni/Ag, and (**h**) etched GaN surface after Cu and Ni/Ag removal.

**Figure 2 nanomaterials-11-03179-f002:**
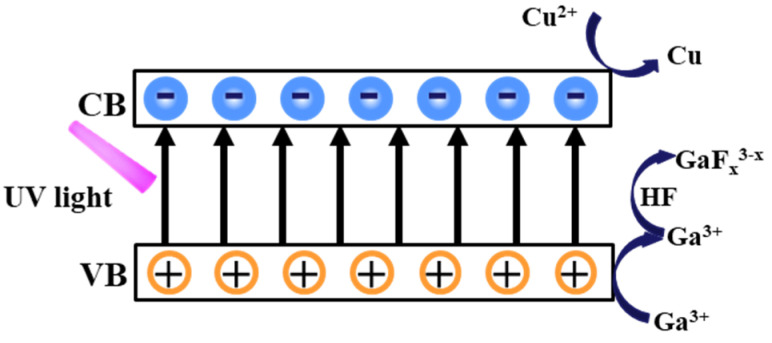
Schematic diagram of the MacEtch mechanism of GaN films.

**Figure 3 nanomaterials-11-03179-f003:**
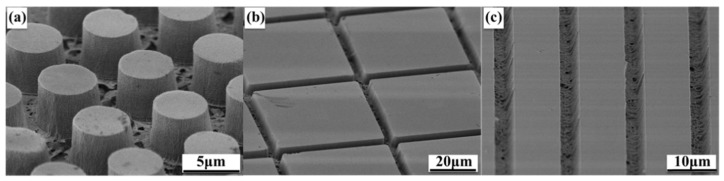
Cross-sectional SEM images (45°-tilt view) of the etched GaN microstructures with different shapes: (**a**) micropillars, (**b**) squares, and (**c**) stripes.

**Figure 4 nanomaterials-11-03179-f004:**
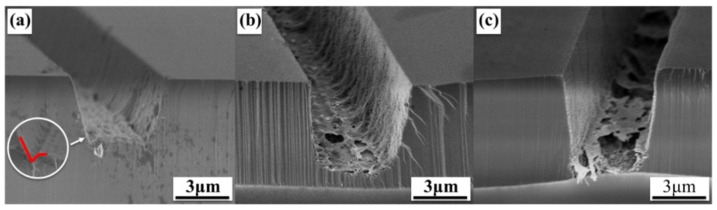
Cross-sectional SEM images (45°-tilt view) of the etched GaN stripe structures with different etching times: (**a**) 20 min, (**b**) 40 min, and (**c**) 60 min.

**Figure 5 nanomaterials-11-03179-f005:**
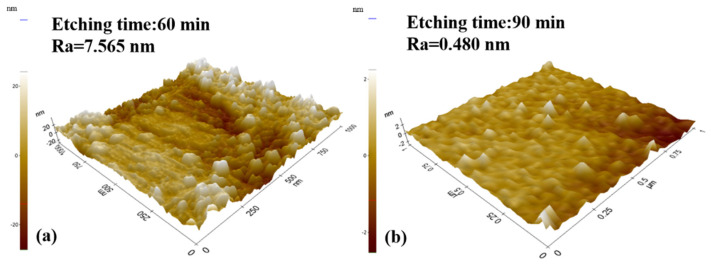
AFM images of the etched surface: (**a**) etching 60 min and (**b**) etching 90 min.

**Figure 6 nanomaterials-11-03179-f006:**
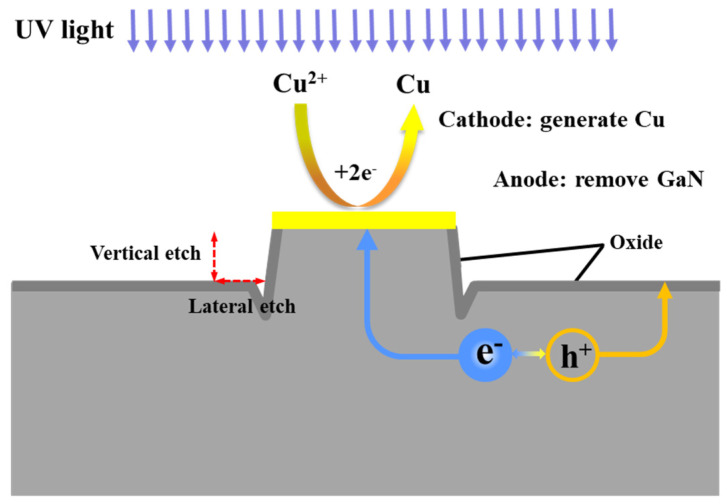
Illustration of the metal-assisted chemical etching mechanism of GaN.

**Figure 7 nanomaterials-11-03179-f007:**
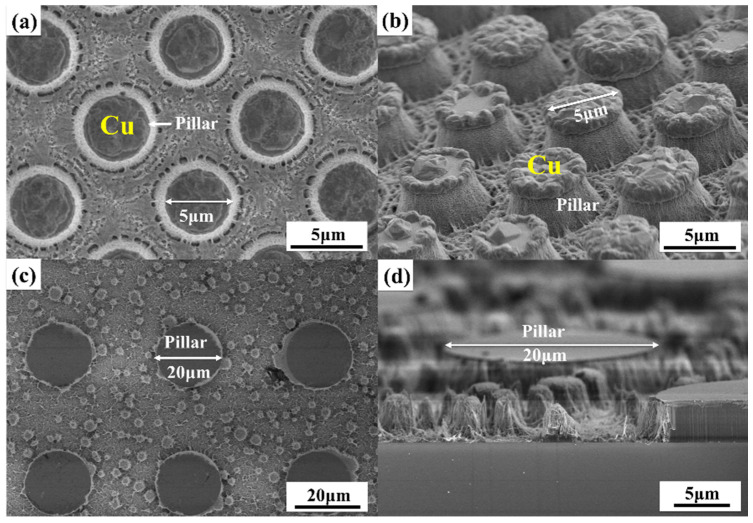
Plane view and cross-sectional SEM images of the obtained GaN microstructures using MacEtch with different mask materials: (**a**,**b**) Ni/Ag mask and (**c**,**d**) SiN mask.

**Figure 8 nanomaterials-11-03179-f008:**
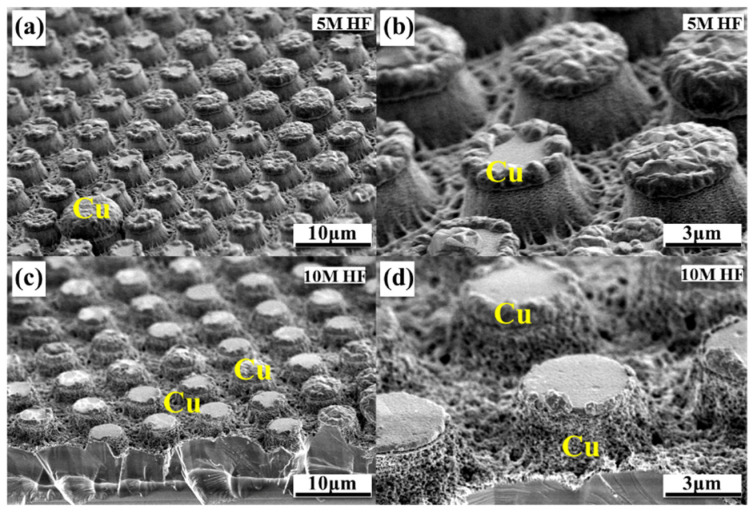
Cross-sectional SEM images (45°-tilt view) of the GaN micropillar arrays obtained with etchants containing different HF concentrations: (**a**,**b**) 5 M and (**c**,**d**) 10 M.

**Figure 9 nanomaterials-11-03179-f009:**
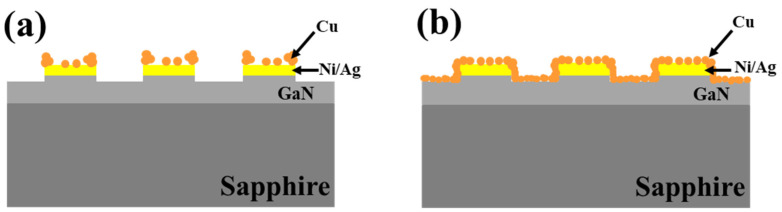
Schematic illustrations for Cu distributions in the GaN MacEtch process with different etchants where the HF concentration is: (**a**) low; (**b**) high.

## Data Availability

All data used to support the findings of this study are included within the article.

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
