# Peer review of "Metal-Assisted Chemical Etching for Anisotropic Deep Trenching of GaN Array"

_nanomaterials, 2021, doi:10.3390/nano11123179_

Round 1
Reviewer 1 Report
Thank you for revising your manuscript accroding to my suggestions.
Author Response
Dear reviewer,
Thank you for your careful read and thoughtful comments. Those comments are valuable and help us to improve the manuscript. We highly appreciate your time and consideration.
Best regards
Dr. Yi Huang
School of Optoelectronic Engineering
Chongqing University of Posts and Telecommunications
Reviewer 2 Report
Dear Editor,
There is considerable improvement in the language in this work, which makes sentences now comprehensible to readers, however, several places for confusion remain. I will cite a few later to clarify my point. In general, I see there is scientific results in the paper worth publishing, but the authors would need to devote more time in the organization and polishing of the paper. I think that readers should not be left the task of figuring out what the authors want to transmit, it is the authors’ obligation to present the information in the clearest way possible and this is not the case yet in this revised version. I also don’t think that Reviewers should point out all the places for improvement. I hope that by citing a few examples or suggestions, the authors can recognize other places (of the multiple existing in this work) where they must improve the message they are giving to the readers.
Everything was going relatively fine for me until section “Selection of Mask Materials.” My few suggestions so far were: 1) Add a schematic and description of the MacEtch process in Section 2, which they do, but later in Section 3. It is hard to follow the process flow before section 3 if the readers are not presented with a simple description of the metal assisted chemical etching principle. More detailed analysis of the process could come later in the manuscript. 2) Rewrite the paragraph in lines 98-108 for clarity. Here, the authors use words such as “shooting angle,” “out of flatness,” and write confusing sentences. It took me a while to figure out what this paragraph, which refers to Fig. 3, means. What they meant, I think, is that the cleaved x-section was not vertical all along the cleavage surface, so they might have had to search regions where x-sections were approximately vertical ending up with different orientations for the groves. And 3) Line 125 has a typo (should be Fig. 2a).
However, starting section “Selection of Mask Materials,” a series of typos and confusing places started to appear at high rate: 1) Authors give for granted that the readers know that Figs. 6a and 6b use a different mask (negative of the one in Fig. 2a). 2) Fig. 6c is tossed into this discussion, without telling the readers that Fig. 6c uses a different mask (pillars), and no discussion is given as of why the authors compare results in holes with results in pillars. 3) Fig. 7 doesn’t describe if it is for a hole or a pillar (readers would again need to figure this out). 4) Yellow color in Fig. 9 doesn’t show. And 5) Line 197 has a typo, it should say “corresponds to Fig. 8d.” Or, maybe not, I am confused.
I would have expected that these kind of confusing places in the manuscript would have been fixed in this version of the manuscript. I reiterate that the list of places for improvement is not limited to the examples I cited. I encourage the authors to try one and last time, however, they should put their best effort in presenting the results in the clearest possible way. I suggest they seek advice from an experienced colleague, who guides them in improving the structure of the paper for clarity and who can detect places of confusion of the kind I listed here.
Author Response
Dear reviewer,
We really appreciate you for your careful read and helpful suggestions. And we have tried our best to improve the manuscript and made some changes in the manuscript according to your comments. The reversions are explained in the response letter and highlighted in the new manuscript. We hope the correction will meet with approval.
Best regards
Dr. Yi Huang
School of Optoelectronic Engineering
Chongqing University of Posts and Telecommunications

Reviewer 3 Report
The paper appears complete and I have no further comments and remarks. I suggest to accept "as is". A very minor point would be only to perform another careful spell-check.
Author Response
Dear reviewer,
Thank you for your valuable comments and helpful suggestions. And we highly appreciate your approval for our work.
Best regards
Dr. Yi Huang
School of Optoelectronic Engineering
Chongqing University of Posts and Telecommunications
Round 2
Reviewer 2 Report
Dear Editor,
I am glad that the authors have substantially improved the presentation of the manuscript. It is obvious that the authors are not native English speakers and the manuscript still has several English style-type errors, like inappropriate use of articles, however, the overall level of the revised manuscript lets readers understand the authors' scientific contributions. I recommend publication after the authors consider a couple of minor comments:
1) In line 61, the authors should consider spelling out “undoped GaN.”
2) In line 91, the authors should spell out the meaning of the symbol used for “oxidation potentials.”
Author Response
Dear reviewer,
We really appreciate you for your affirmation and comment on our modification. And we have revised the manuscript according to your comments. The reversions are explained in the response letter and highlighted in the new manuscript. We hope the correction will meet with approval.
Best regards
Dr. Yi Huang
School of Optoelectronic Engineering
Chongqing University of Posts and Telecommunications

This manuscript is a resubmission of an earlier submission. The following is a list of the peer review reports and author responses from that submission.
Round 1
Reviewer 1 Report
This paper describes metal-assisted chemical etching method for GaN. This method works well and obtained results are reasonable and fruitful. There are some questions are remained. Please comment or check them.
- In title, “damage-free” is confusion because usually “damage-free” uses electronic property. In this paper, electronic properties of etched GaN did not measure. So, in this case, “smooth (or good shape? etc.) deep trenching” is suitable.
- Line 73, “lithography”, Please explain this process in detail.
- Typos: Line 17, CuSO4 -> 4 is subscript. Line 100, poles -> holes? Please check it.
Reviewer 2 Report
While it seems like the author is attempting to propose a new etching method, the connections between the contents of the paper and the etching method are not clearly stated. The figures often lacked explanations as to which part of the experiment they are referring to or why certain tasks were performed. The figures were also insufficiently labelled, and the captions were of insufficient lengths to describe any meaningful connection to the main text. The lack of any quantitative analysis also makes it difficult to believe that the proposed etching method is sufficient in its applications. Written below are some examples.
- Lack of details and labels
In figure 1, chronology is nowhere to be found. How did you obtain the pattern in figures 1(b) ~ 1(d)? Do the figures 1(b) ~ 1(d) depict the states of the sample before or after being placed in the etchant? As the first figure of the paper, a reader may expect to find a schematic of the overall process, or at least something written in the captions. It is also difficult to find further explanations for the steps depicted in figure 1 anywhere in the text. For instance, why did you deposit Ni/Ag in figure 1(b)?
What are the materials in figures 8 (b) and (c) (the yellow and the orange ones)? Not even written in the captions.
- Lack of quantitative analysis
SEM captures are means of qualitative analyses- they often lack abilities in the quantitative aspects. For instance, figures 2(a) and (b) both contain residues that seem to render them unusable for many applications. Despite one sample having a cleaner surface than the other, both samples have rough surfaces and without quantitative analysis, it is difficult to draw comparisons between the two figures. The sample with a longer etching time also has a rougher surface- what would you need to do to get a clean surface? Any optical/electrical measurements to back-up your claims?
None of the SEM captures in the paper show a “cleanly etched” surface. If you were to make a claim that the new etching method you are proposing is an effective one, it may be useful to either (i) add a picture that is actually clean, instead of samples with rough surfaces; or (ii) add quantitative analyses to suggest that samples fabricated with your method made the quality better.
- Lack of coherence within figure sets
Figure 2(c) seems to describe a completely different sample than in 2(a) or 2(b). Why is it there? What point are you trying to make?
- Language
The quality of written communication in this paper makes the aforementioned issues even more devastating to the overall flow of the paper.
Perhaps I’m being too nit-picky at this point, but there must be a space between a number and its units (e.g., it’s 20 min not 20min, 200 nm not 200nm). It’s also inconsistent throughout the paper.
Reviewer 3 Report
I am sorry to say that the quality of English is not appropriate for a scientific publication. Therefore I don't want to spend my time to review the manuscript, it needs considerable language editing. Please come back with a better manuscript, maybe get some professional help.
Reviewer 4 Report
Dear Editor, This work on chemical etching of GaN is hard to follow for me as a reviewer who is not specialized in the paper's subject but who has significant experience in nanofabrication of electronic devices. Consequently, I foresee that the manuscript will be very hard to follow for an audience with broad interest in nanotechnology. I recommend the authors to seek publication in a specialized (chemistry leaning) journal in which the audience is already familiar with the topic. Otherwise, to publish in this journal, the authors would have to do a major revision of the manuscript to introduce the technique in an effective way, and to avoid confusions that are scattered in several places of the current manuscript. I’ll cite a couple of examples to make my point. In Section 2, the authors assume that readers are already familiar with the technique. The need for UV light to generate hole-pairs, the reaction mechanism, and the fact that Cu is transferred onto the Ni/Ag pattern surface is left to the reader to figure out by themselves. The authors do not inform the reader that these facts will be explained in detail later in the manuscript. The authors assume that this knowledge is evident to readers, but I consider it as a source of confusion. Figure 2 (SEM images) do not correspond to schematic in Fig. 1. According to Fig. 1, the reader expects to see pillars, while figures 2a and 2b show holes (authors have the typo “poles”). What happened here? Did the authors change the metal mask to the negative version shown in Fig. 1? This is also left to the readers to figure out. Also, the roughness of the features shown in Fig. 2 contradict the “damage-free” adjective used in the title of the manuscript. The authors do not comment on the origin of the observed roughness at this point in the manuscript.